# Isopeptidase Kinetics Determination by a Real Time and Sensitive qFRET Approach

**DOI:** 10.3390/biom11050673

**Published:** 2021-04-30

**Authors:** Yan Liu, Yali Shen, Yang Song, Lei Xu, J. Jefferson P. Perry, Jiayu Liao

**Affiliations:** 1Department of Bioengineering, Bourns College of Engineering, University of California at Riverside, 900 University Avenue, Riverside, CA 92521, USA; yanliu0405@gmail.com (Y.L.); songy99@hotmail.com (Y.S.); 2Department of Abdominal Oncology, Cancer Center, West China Hospital, Sichuan University, Chengdu 610041, China; sylprecious@163.com; 3Department of Geography & the Environment, California State University, Fullerton, 800 N State College Blvd, Fullerton, CA 92831, USA; lxu@fullerton.edu; 4Department of Biochemistry, University of California at Riverside, 900 University Avenue, Riverside, CA 92521, USA

**Keywords:** isopeptidase, enzyme kinetics, quantitative FRET (qFRET) assay, SUMOylation, SENP

## Abstract

Isopeptidase activity of proteases plays critical roles in physiological and pathological processes in living organisms, such as protein stability in cancers and protein activity in infectious diseases. However, the kinetics of protease isopeptidase activity has not been explored before due to a lack of methodology. Here, we report the development of novel qFRET-based protease assay for characterizing the isopeptidase kinetics of SENP1. The reversible process of SUMOylation in vivo requires an enzymatic cascade that includes E1, E2, and E3 enzymes and Sentrin/SUMO-specific proteases (SENPs), which can act either as endopeptidases that process the pre-SUMO before its conjugation, or as isopeptidases to deconjugate SUMO from its target substrate. We first produced the isopeptidase substrate of CyPet-SUMO1/YPet-RanGAP1c by SUMOylation reaction in the presence of SUMO E1 and E2 enzymes. Then a qFRET analyses of real-time FRET signal reduction of the conjugated substrate of CyPet-SUMO1/YPet-RanGAP1c to free CyPet-SUMO1 and YPet-RanGAP1c by the SENP1 were able to obtain the kinetic parameters, *K_cat_*, *K_M_*, and catalytic efficiency (*K_cat_/K_M_*) of SENP1. This represents a pioneer effort in isopeptidase kinetics determination. Importantly, the general methodology of qFRET-based protease isopeptidase kinetic determination can also be applied to other proteases.

## 1. Introduction

Reversible post-translational modification of proteins by small chemical groups, sugars, lipids, and polypeptides is an important means to alter their function, activity, or localization after their synthesis have been completed [1,2]. SUMO (small ubiquitin-like modifier) covalently modifies and regulates the activities of proteins that have important roles in diverse cellular processes, including cell cycle regulation, cell survival and apoptosis, DNA damage responses, and stress responses [2,3,4,5,6].

Like ubiquitination, SUMO conjugation occurs through a cascade of reactions performed by an activating enzyme (E1), a conjugating enzyme (E2) and, usually, a SUMO ligase (E3). RanGAP1 was the first identified SUMO target, and its SUMOylation cycle mediates the constant shuttling of this protein between the cytoplasm and nucleus [7]. The formation of an isopeptide bond between SUMO and Lys526 of RanGAP1 in vitro occurs in the presence of E1, E2 enzyme, and ATP. This reaction does not require E3 ligase RanBP2 [7], as RanBP2 can function in coordinating the SUMO-Ubc9 thioester in an optimal conformation for catalysis to enhance SUMOylation, but without directly contacting the RanGAP. The SENPs (sentrin-specific proteases) perform two critical functions via their cysteinyl proteinase activity. The first involves proteolysis of SUMO C-terminal amino acid residues, so as to release a mature form of SUMO protein that is terminated with a di-Gly motif. This SENP-processed form of SUMO is the only known form that can be activated and conjugated to other proteins. The second protease activity of SENPs catalyzes SUMO deconjugation from a SUMOylated target protein, releasing both the target lysine and SUMO [8]. Seven SENPs have been identified in the human genome, comprising SENP1-3 and 5-8, and these SENPs all contain a conserved ~200 amino acid C-terminal catalytic domain that has the SUMO cleaving activity. The N-terminal regions of the SENPs are unique to other proteins and appear to have functions in localizing SENPs to different locations of the cell. SENP8 is an outlier within this family, as it is not a SUMO protease per se but instead functions on another small Ubl that is known as Nedd8. [8,9]. SENPs participate in diverse biological pathways, including transcriptional regulation, development, cell growth and differentiation, cancer, and ribosome biogenesis [10]. Different SENPs demonstrate various specificities toward SUMO substrates, and the isopeptidase and endopeptidase activities of one SENP to the same SUMO substrate may also not be the same.

The catalytic efficiency or specificity of an enzyme is best characterized by the ratio of the kinetic constants, *Kcat/K_M_*. Several methods are commonly used to determine *Kcat/K_M_*, such as the enzymatic digestion in solution, followed by the polyacrylamide gel based Western blot method, radioactive-labeled substrate, dialysis of digested substrate, fluorescent compound-labeled peptide substrate or fluorescent protein-labeled substrate. The activities of SENPs to process pre-SUMOs, and remove SUMOs from RanGAP1, have been characterized by Western blotting [11,12]. Some studies used tetrapeptides with di-Gly motif and labeled an organic fluorophore ACC (7-amino-4-carbamoylmethylcooumarin), which can emit fluorescence signal when cleaved by SENPs. The range of the determined *Kcat/K_M_* values was 17–325 M^−1^·s^−1^, which was up to two orders of magnitude lower than that using the natural substrates, as the SUMO substrates differ outside of the catalytic cleft (di-Gly motif) have significant impact on the binding step (*K_M_*), therefore affecting its kinetics activities [12,13]. To compare the specificities of different SENP paralogues and to measure the kinetics parameters, mature SUMO1/2 were tagged with a similar organic fluorophore, AMC (7-amino-4-methylcoumarin). The determined *K_cat_/K_M_* value was 2.4 × 10^6^ M^−1^·s^−1^ for SENP1-SUMO1, but there is no peptide of SUMO tail, and therefore the applied system cannot clearly demonstrate the iso- or endo-peptide function of SENPs [14].

Förster resonance energy transfer (FRET) is widely used in biological and biomedical research, including cell biology, medical diagnostics, optical imaging, and drug discovery [15,16,17,18]. FRET occurs when the donor fluorophore (D) and acceptor fluorophore (A) are close to each other (1 to 10 nm) with favorable orientations. The excitation of the donor can elicit an energy transfer to induce emission from the acceptor, which results in quenching of donor and excitation of acceptor. Fluorescent proteins are being increasingly used in FRET systems, due to the ease of genetic labeling of protein of interest with a fluorescent tag. Ease of tagging coupled with the advantages of a high sensitivity and a suitability of FRET-assays for both spectroscopy and imaging analysis, has led to FRET-based proteases assays being used to study both deubiquitinating enzymes (DUBs) and SENPs. One FRET pair, Eu-cryptate and APC (allophycocyanin), was tagged to anti-Myc and anti-FLAG antibodies, respectively, which interacted with Myc and FLAG on the N- or C-terminus of tagged Myc-pre-Nedd8-FLAG. Terbium (Tb) and YFP (yellow fluorescent protein), form another FRET pair, and these were previously tagged on SUMO and anti-RanGAP, respectively, to study the SUMOylation and SENP’s deconjugation. The Tb/YFP FRET pair was also used on N- and C- terminus of ubiquitin, so as to study the DUB processing by time-resolved FRET (TR-FRET) technology [19,20,21,22,23,24]. ECFP (enhanced cyan fluorescent protein) and YFP were also used as the FRET pair to study SENP1’s activities [20].

Yet, these FRET-based assays still require additional steps for their detections, which includes immune antibodies conjugation or chemical conjugation of thiol-reactive Tb chelate to ubiquitin-AC or other fluorophores. Both the conjugation efficiency and the indirect measurement may therefore lead to inaccurate results during the quantitative analysis. However, all of the above FRET-based protease assays used the ratio of acceptor’s emission to donor’s emission (under the excitation of donor) to characterize the FRET signals, and thus self-fluorescence from donor and acceptor that can lead to an inaccurate FRET signal analysis was not considered [25,26]. Furthermore, the complexity of fluorescence emissions of the undigested-, digested-substrate as well as free donor, acceptor at the emission wavelength of acceptor limit assay reliability and sensitivity.

We have recently developed the highly sensitive and qFRET-based protease assay, so as to characterize SENPs’ endopeptidase kinetics [27,28,29,30]. We used an engineered FRET pair, CyPet and YPet that have a significantly improved FRET efficiency and fluorescent quantum yield, to generate the CyPet-(pre-SUMO)-YPet substrate [27,28]. In these methods, we developed a novel quantitative FRET analysis method to extract the absolute FRET signal, which corresponds to un-digested substrate, CyPet-(pre-SUMO)-YPet, from the fluorescence signals directly from free donor, CyPet-(pre-SUMO), or acceptor, -YPet, from digested substrate. We used either an external standard curve of free donor CyPet, acceptor YPet, and FRET substrate, CyPet-(pre-SUMO)-YPet or an internal cross-correlation coefficiency method [28,29,30]. Our approach can determine accurate digested or undigested substrates without interferences of free donor and acceptor in real time. Here, we use a novel strategy, which now enables us to determine SENP isopeptidase kinetics. In this design, CyPet and YPet were genetically fused to SUMO1/2 or the C-term domain of RanGAP1 (420–587), respectively, to generate CyPet-SUMO1/2-RanGAP1c-YPet substrate in the presence of SUMOylation E1 and E2 enzymes. We differentiated and quantified the absolute fluorescent signals contributed by the donor, the acceptor, and FRET at the acceptor’s emission wavelengths, respectively, during the process of SUMO deconjugation by the SENP1. In addition to using standard curves of undigested substrate and free fluorescence proteins to correlate the fluorescent reading to the amount of hydrolyzed substrate, we also used the real-time fluorescent reading at both donor and acceptor’s emission wavelength using an internal control references [28]. We obtained a value of *K_cat_/K_M_* of SENP1 to deconjugate SUMO1 from RanGAP1c through our qFRET analysis as (4.35 ± 1.46) × 10^7^ M^−1^·s^−1^. This represents the first effort in determining a true isopeptidase kinetics.

## 2. Materials and Methods

### 2.1. Plasmid Constructs

The open reading frames of the genes were amplified by PCR, and the PCR products were cloned into PCRII-TOPO vector (Invitrogen). After confirming the constructs by sequencing, the cDNAs encoding CyPet-SUMO1/2, YPet, Aos1, Uba2, Ubc9, and the catalytic domains of SENP1/2/5/6/7 [12] were cloned into the pET28 (b) vector (Novagen), engineered with an N-terminal poly-histidine tag. The cDNA encoding YPet-RanGAP1C was cloned into the pGEX4T-1 vector (GE Healthcare) with an N-terminal GST-tag.

### 2.2. Protein Expression and Purification

*Escherichia coli* cells of strain BL21(DE3) were transformed with pET28 vectors encoding CyPet-SUMO1/2, YPet, Aos1, Uba2, Ubc9 and the catalytic domains of SENP1/2/5/6/7 as well as with pGEX4T-1 vector encoding YPet-RanGAP1C. The transformed bacteria were grown in 2xYT medium to an optical density at 600 nm of 0.4–0.5, by induction with 100 µM isopropyl-β-D-thiogalactoside (IPTG) for 16 h at 25 °C. The poly-histidine-tagged recombinant proteins were purified from bacterial lysates with nickel agarose affinity chromatography (QIAGEN, Hilden, Germany) and eluted with 500 mM imidazole. The GST-tagged recombinant proteins were purified from bacterial lysates with glutathione agarose affinity chromatography (Thermo Scientific, Waltham, MA, USA) and eluted with 20 mM reduced glutathione. All the recombinant proteins were dialyzed in 20 mM Tris-HCl, pH7.4, 50 mM NaCl, 1 mM DTT. Protein purity was examined by SDS-PAGE, and concentrations of the purified proteins were determined by Bradford Assay (Thermo Scientific, Waltham, MA, USA).

### 2.3. Preparation of SUMOylated RanGAP1C Substrate

Total of 7.2 mg GST-YPet-RanGAP1C was conjugated to 4 mg His-CyPet-SUMO1/2 in 30 mL reactions with 0.5 mg Aos1, 1 mg Uba2, 5 mg Ubc9 in the buffer containing 50 mM Tris-HCl, 150 mM NaCl, 5 mM MgCl2, 5 mM DTT, 0.1% BSA. 0.1% (*v*/*v*) Tween-20 and 5 mM ATP. The reactions were performed in 37 °C up to 2 h.

The SUMOylated RanGAP1C was then purified by glutathione-GST affinity chromatography, followed by nickel-6xHis affinity chromatography, and dialyzed overnight in dialysis buffer (20 mM Tris-HCl, 50 mM NaCl, 1 mM DTT, pH 7.4).

The purity of the proteins was confirmed by SDS-PAGE, and concentrations of protein were determined by Assay with known quantities of bovine serum albumin as standards. Aliquots of final products were stored in −80 °C.

### 2.4. Protease Assay to Study the Specificities

FRET-based SUMO deconjugation assays were conducted by measuring the emission intensity of CyPet at 475 nm and YPet at 530 nm with an excitation wavelength of 414 nm in a fluorescence multiwall plate reader FlexStation II384 (Molecular Devices, Sunnyvale, CA, USA).

To test the substrate specificities, CyPet-SUMO1/2-RanGAP1C-YPet-GST was incubated with catalytic domains of SENP1/2/5/6/7 (1:1 molar ratio) at 37 °C in low salt reaction buffer (20 mM Tris-HCl (pH 7.4), 50 mM NaCl, 0.1% (*v*/*v*) Tween-20 and 1 mM DTT) and transferred into a 384-well plate (glass bottom, Greiner, Monroe, NC, USA). The final concentration of reacted proteins was 100 nM. Reactions were stopped at 1 hr and were analyzed by fluorometer. Three samples were repeated in each condition. The results were reported as mean ± SD.

To study the differences of endopeptidase and isopeptidase activities of SENP1, CyPet-(pre-SUMO1/2)-YPet and CyPet-SUMO1/2-RanGAP1C-YPet-GST were incubated with SENP1C separately at 37 °C in low salt reaction buffer and transferred into a 384-well plate. The final concentrations of substrates and enzymes were 100 nM and 0.5 nM respectively. Reactions were tested within original 5 min with 10 s intervals. Initial velocities were derived by the developed method. Five samples were repeated in each concentration.

### 2.5. Self-Fluorescence Cross-Talk Ratio Determination

To determine the cross-talk ratio of CyPet and YPet self-fluorescence, purified CyPet-SUMO1 and YPet were incubated individually in 37 °C in buffer containing 20 mM Tris-HCl (pH 7.4), 50 mM NaCl, 0.1% (*v*/*v*) Tween-20 and 1 mM DTT to a total volume of 80 µL in the concentration of 10 nM, 20 nM, 50 nM, 100 nM, 200 nM, and 500 nM for 10 min and added to each well of a 384-well plate (glass bottom, Greiner, Monroe, USA).

Fluorescent missions of CyPet at 475 nm and 530 nm were detected in a fluorescence multi-well plate reader (Molecular Devices, Flexstation II384) under the excitation at 414 nm to determine the cross-talk ratio *α*; fluorescent emissions of YPet at 530 nm were detected under the excitation at 414 nm and 475 nm to determine the cross-talk ratio *β*. Three samples were repeated for each concentration.

### 2.6. Protease Kinetics Assay

FRET-based SUMO processing assays were conducted by measuring the emission intensity of CyPet at 475 nm and of YPet at 530 nm with an excitation wavelength of 414 nm in a fluorescence multi-well plate reader (Flexstation II^384^, Molecular Devices, San Jose, CA, USA).

Recombinant CyPet-SUMO1-RanGAP1C-YPet substrate with different concentrations were incubated with recombinant 0.267 nM catalytic domain of SENP1 at 37 °C in buffer containing 20 mM Tris-HCl (pH 7.4), 50 mM NaCl, 0.1% (*v*/*v*) Tween-20, and 1 mM DTT to a total volume of 80 µL and added to each well of a 384-well plate (glass bottom, Greiner, Monroe, MI, USA). Reactions were tested within original 5 min. One phase association model was used to fit the exponential increased reaction velocity. Data were analyzed by the developed method and plotted in GraphPad Prism V software fitting the Michaelis–Menten equation. Five samples were repeated in each concentration.

## 3. Results

### 3.1. Design of FRET-Based Isopeptidase Activity Determination and SUMOylated Substrate RanGAP1-SUMO Preparation

RanGAP1 was the first identified SUMO target, and its SUMOylation cycle mediates the constant shuttling of this protein between the cytoplasm and nucleus [7]. The formation of an isopeptide bond between SUMO and Lys526 of RanGAP1 in vitro occurs in the presence of E1, E2 enzyme and ATP, and without the assistance of the E3 ligase.

Similar to the design of the FRET-based protease assay to study the endopeptidase activity of SENPs [27,28,30], the FRET pair CyPet and YPet was applied to monitor the SUMOylation of RanGAP1, and also the deconjugation SUMO from the SUMO-RanGAP substrate by SENPs. CyPet and YPet were genetically tagged to the N-terminus of SUMO1/2 and RanGAP1c respectively. With the existence of SUMO E1 (Aos1/Uba2), E2 enzyme (Ubc9) and ATP, CyPet-SUMO1/2 will be covalently linked to YPet-RanGAP1c. The excitation of CyPet will transfer energy to the YPet that is in close proximity, and the resultant quenching of donor and increased emission of acceptor can be observed. When CyPet-SUMO-RanGAP1c-YPet is mixed with SENP, it is cleaved by the protease, resulting in two products: the CyPet–SUMO and the YPet-RanGAP1c. Therefore, the FRET signal will be disrupted, resulting in an increase of CyPet’s emission, as well as a dramatic decrease of YPet’s emission when CyPet has been excited; this change in emissions can be used to characterize kinetic properties of SENP1 in real time (Figure 1).

CyPet-SUMO1/2 and YPet-RanGAP1C proteins were first mixed with Aos1, Uba2, and Ubc9 proteins in the low-salt Tris buffer (50 mM Tris-HCl, 150 mM NaCl, 5 mM MgCl2, 5 mM DTT, 0.1% BSA. 0.1% (*v*/*v*) Tween-20 and 5 mM ATP) without ATP. ATP was added at time point 0, and the fluorescent emission at 475 nm and 530 nm as well as the emission ratio (*FL530/FL475*) of the protein mixture were monitored every 2 min. Compared with the negative control sample that did not have ATP, the sample with ATP presented a significant decrease of fluorescence emission at 475 nm and an increase of fluorescence emission at 530 nm. The emission ratios (*FL530/FL475*) were also dramatically increased, and reached the plateau in 30 min (Figure 2a,b).

The multiple Lys residues in the N-terminal extensions of SUMO2 can form polySUMO chains, while SUMO1 cannot and is incapable of extending a SUMO chain [31]. SENP6 and SENP7 have been discovered to have the ability to edit polySUMO tail [32]. The molar ratio of CyPet-SUMO2 to YPet-RanGAP1c used in the SUMOylation assay was about 1:1, which was expected to modify every RanGAP1c by SUMO while avoiding the production of a polySUMO chain on RanGAP1c. As all the SUMO2 were genetically tagged with CyPet, the formation of polySUMO2 chains would lead to the multiple SUMO2 on one RanGAP1c, or in other words, multiple CyPet with one YPet. The detected fluorescent emission of 475 nm and increased fluorescent emission of 530 nm should not be the same as the emissions for one donor coupled one acceptor in the FRET system. The fluorescent emission changes at 475 nm and 530 nm, as well as the emission ratio of SUMO2-RanGAP1c conjugation, were the same as those of SUMO1-RanGAP1c conjugation. This denoted that there was no formation of polySUMO2 chains on RanGAP1c in the protein assay under the experimental conditions used (Figure 2a,b).

To obtain the pure substrate of the protease kinetics study, different tag affinity purification methods of 6xHis-Ni and GST (glutathione S-transferase)–glutathione were used. The GST sequence was incorporated into the bacterial expression vector (pGEX4T-1, GE-Healthcare) alongside the gene sequence encoding YPet-RanGAP1C. Induction of protein expression from the tac promoter resulted in the expression of the fusion protein GST-YPet-RanGAP1C. The GST tag has the size of 220 amino acids (~26kDa), which, compared to other tags like the Myc- or the FLAG-tag, is relatively large, which may interfere with the protease-substrate interaction. Agarose beads coated with glutathione, the GST substrate, can bind to GST-fused YPet-RanGAP1C.

All the recombinant proteins were tagged with 6xHis, except YPet- RanGAP1C. The reaction system of SUMOylation included CyPe-SUMO1/2, Aos1, Uba2, Ubc9, and YPet-RanGAP1C. After SUMOylation, the reaction system first flows through agarose beads coated with glutathione to bind CyPet-SUMO1/2-RanGAP1C-YPet and un-SUMOylated YPet-RanGAP1C, and to remove by elution the SUMO E1, E2 enzyme and the unused CyPet-SUMO1/2. The eluted proteins were then run through a Ni-NTA agarose bead solution, to bind the CyPet- SUMO1/2-RanGAP1C-YPet, and to remove the unSUMOylated YPet-RanGAP1C. The SUMOylated RanGAP1C was eluted by imidazole, and dialyzed over night to remove the excess salt. The same size of CyPet-SUMO1-RanGAP1C-YPet-GST and CyPet-SUMO2-RanGAPC-YPet-GST observed on the polyacrylamide gel confirmed that there were no polySUMO2 chains conjugated to RanGAP1C (Figure 3).

The SUMOylated RanGAP1C was then incubated with different SENPs in 1:1 molar ratio at 37 °C for 1 hr. The fluorescent emission ratio (*FL530/FL475*) under the excitation of 414 nm was used to characterize the changes of FRET signals (Figure 4).

The results indicated that different SENPs exhibit various specificities towards SUMO deconjugation: both SENP1C and SENP2C can deconjugate SUMO1 and SUMO2 from target substrate, RanGAP1C; SENP5, SENP6 and SENP7 prefer SUMO2 in SUMO deconjugation, and exhibit poor activities toward SUMO1 deconjugation, all of which is in agreement with previous observations [20,32,33,34].

### 3.2. QFRET Analysis and Determination of FRET Signal

The cross-talk ratio of CyPet self-fluorescence (*α*) is the ratio of CyPet-SUMO1’s emission at 530 nm (*I_d_*_530/414_) to 475 nm (*I_d_*_475/414_) under excitation at 414 nm (Figure 5a).
(1)$$\alpha =\frac{I_{d530/414}}{I_{d475/414}}$$

The determined value of *α* is 0.332.

The cross-talk ratio of YPet self-fluorescence (*β*) is the ratio of YPet emission at 530 nm under excitation at 414 nm (*I_a_*_530/414_) to emission at 530 nm under excitation at 475 nm (*I_a_*_530/475_) (Figure 5b).
(2)$$\beta =\frac{I_{a530/414}}{I_{a530/475}}$$

The determined value of *β* is 0.026.

In this way, we can distinguish the detected fluorescence signal at 530 nm under the excitation at 414 nm (*FL*_530/414_) into three fragments: FRET-induced acceptor’s emission (*I_da_*), donor’s direct emission (*I_d_*_530/414_) and acceptor’s direct emission (*I_a_*_530/414_) (Figure 5c)
(3)$$FL_{530/414}=I_{da}+I_{d530/414}+I_{a530/414}$$

According to the determined *α* and *β*:(4)$$FL_{530/414}=I_{da}+\alpha I_{d475/414}+\beta I_{a530/475}$$
where *I_d_*_475/414_ is CyPet emission at 475 nm under excitation at 414 nm, *I_a_*_530/475_ is YPet emission at 530 nm under excitation at 475 nm.

After the hydrolysis by SENP1, the fluorescent signal at 530 nm was decreased and fluorescent signal at 475 nm was increased from the disruption of FRET. The remaining fluorescent emission at 530 nm (*FL*’_530/414_) can still be divided into the same three parts:(5)$$FL_{530/414}^{\prime }=I_{da}^{\prime }+\alpha I_{d475/414}^{\prime }+\beta I_{a530/475}^{\prime }$$
where *I’_da_* is the remaining FRET-induced acceptor’s emission, *I*’_d475/414_ is the fluorescent emission of CyPet that can divided into two parts: from the undigested CyPet-SUMO1-RanGAP1C-YPet and from digested CyPet-SUMO1, *I*’_a530/475_ is the fluorescent emission of YPet, which is constant no matter substrate has been digested or not.

After treatment with SENP1, the remaining FRET-induced acceptor’s emission (*I’_da_*) is:(6)$$\frac{C-x}{C}\times I_{da}=\frac{C-x}{C}\times \left(FL_{530/414}-\alpha I_{d475/414}-\beta I_{a530/475}\right)$$
where *C* is the concentration of CyPet-SUMO1-RanGAP1C-YPet (µM) in 80 µL; *x* is the concentration of digested CyPet-SUMO1-RanGAP1C-YPet (µM) in 80 µL.

In this way, the detected fluorescent signal at 530 nm under excitation at 414 nm can be expressed as:(7)$$FL_{530/414}^{\prime }=\frac{C-x}{C}\times \left(FL_{530/414}-\alpha I_{d475/414}-\beta I_{a530/475}\right)+\alpha I_{d475/414}^{\prime }+\beta I_{a530/475}$$

### 3.3. Initial Velocity Determination of Isopeptidase and Enzyme Kinetics Determination

The SUMO deconjugation by SENPs can be determined by monitoring the changes of fluorescent signal at 475 nm and 530 nm under excitation at 414 nm during the deconjugation process. The concentration of digested substrate, *x*, was calculated according to the above analysis. The initial velocity (*V_o_*) of SUMO deconjugation by SENP was determined using methods previously described as:(8)$$V_{o}=\frac{d\left[P\right]}{dt}{|}_{t=0}=k{\left[S\right]}_{o}$$

The catalytic specificity and efficiency of an enzyme for a specific substrate is best defined by the ratio of the kinetic constant, *K_cat_/K_M_*. This ratio is generally used to compare the efficiencies of different enzymes with one substrate, or the use of different substrates by a particular enzyme.

The *K_M_* and *V_max_* values can be obtained from the Michaelis–Menten equation by plotting the various velocities of SENP1 digestion versus the corresponding different concentrations of substrate. The obtained initial velocities were plotted in Michaelis–Menten model.

### 3.4. QFRET Analysis in Protease Kinetic Study of SUMO Deconjugation

SENP can function as endopeptidase to mature SUMO precursor, or as isopeptide to deconjugate SUMO from its target protein. As both SENP1C and SENP2C can mature pre-SUMO1/2/3 or deconjugate SUMO1/2 from RanGAP1C during a long-term hydrolysis study, the kinetics of the two processes have both to be studied to compare the different activities of SENPs.

To investigate the differences of endo- and iso-peptidase activities of SENP1C, the same concentration (100 nM) of CyPet-(pre-SUMO1/2)-YPet and CyPet-SUMO1/2-RanGAP1C-YPet-GST were incubated with SENP1C (0.5 nM) at a 200:1 molar ratio respectively. The substrate digestions were monitored over 5 min with 10 s intervals (Figure 6) and the initial velocities under ([S] = 100 nM) were derived by the above qFRET analysis (Table 1). The results indicated that SENP1 exhibited higher activity toward SUMO deconjugation than pre-SUMO maturation (especially for SUMO2).

Interestingly, the preferences were different for the same SUMO-SENP pair. In pre-SUMO maturation, SENP1 preferred pre-SUMO1 than pre-SUMO2, but in the process of SUMO deconjugation, SUMO2 was preferred by SENP1 than SUMO1.

As analyzed above, the change of fluorescent emission of 475 nm and 530 nm (under the excitation of 414 nm) can be used to characterize the kinetic process of SENP1C hydrolysis toward its substrate (CyPet-SUMO1-RanGAP1C-YPet).

The developed qFRET analysis was used here to derive the initial velocities under different substrate concentrations (Figure 7a and listed in Table 2). The kinetic constants were derived by plotting the initial velocities versus substrate concentration in Michaelis–Menten equation (Figure 7b) and listed in Table 3. The values of derived kinetic constants were close to the previous studies [20].

## 4. Discussion

The accurate determinations of kinetics parameters are critical for estimating enzymatic activity, specificity, and/or drug candidate evaluation. A variety of FRET-based protease assays have been developed to study the protease kinetics. For example, those methods include a Lanthanide assay that was combined with TR-FRET technology and the genetically tagged ECFP-YFP on the SENP protein of interest [15,20,21,22,35]. All the previous studies determine endopeptidase kinetics parameters. So far, no isopeptidase kinetics parameter has been reported. We, for the first time, developed a FRET-based methodology to determine the protease isopeptidase kinetics using a CyPet-SUMO1-conjugated substrate, CyPet-SUMO1-YPet-RanGAPc, after double purifications of Ni and Glutathione beads sequentially. We then determined the amount of digested substrate using our qFRET method. This methodology represents the first effort in isopeptidase kinetics determination.

In our systematical effort of developing a qFRET universal assay methodology to determine all kinetics of SUMOylation cascade and others, we have focused on developing a qFRET analysis method for various biochemical reactions. Unlike previous ratiometric FRET analyses, our qFRET analysis considered the direct emissions of donor and acceptor at acceptor’s emission wavelength. The pre-established standard curve in our initial approach for SENP endopeptidase kinetics determination can correlate the concentrations of substrate and product to detected fluorescent signals [28]. We later improved the standard curve-dependent qFRET analysis, to quantify the FRET signal in internal calibration [30]. The emission of FRET donor CyPet can be directly obtained by the spectrometer. Here, we applied both standard curve and the internal calibration of FRET quantification methods, to characterize the isopeptidase specificities and kinetics of SENP1. Compared to the previous standard curve method, which the fluorescent signal was in linear curve to fit the protein concentrations of digested substrate, the quantification of FRET by improved internal calibration method has more advantages in minimizing the variations from either the spectrometer or the protein sample preparation.

A little appreciable difference between structure complex pre-SUMO1-SENP1 (C603A) and SENP1 (C603A)-RanGAP1-SUMO1 has been observed with the exception that the side chain of His533 (part of the catalytic triad) has an altered conformation [20]. The activities of SENP1 in maturing pre-SUMO1, or in deconjugating SUMO1 from RanGAP1, had very similar values of initial velocity under the same substrate concentration, resulting in $K_{M}$ and $k_{cat}$ values for maturation or deconjugation being close to each other (Figure 6, Table 1).

As compared to the traditional ratiometric FRET analysis, we improved the FRET approach in both providing a new theory of FRET signal for kinetic analysis and in developing an experimental procedure to derive kinetic parameters. These parameters are derived from the quantitative contributions of absolute fluorescent signals from donor’s direct emission, acceptor’s direct emission and real FRET-induced acceptor’s emission. The small numeric differences observed between the developed qFRET analysis and ratiometric analysis may reflect a fundamental difference of the FRET data processing. The discrepancies between these two approaches might be due to the inclusion of direct emission of donor and acceptor in the ratiometric analysis method. Such an inclusion, resulting in overestimations of FRET signal, might not greatly affect the final *K_cat_/K_M_* ratio, but the signal contamination effect likely becomes more evident when studying the individual parameters of *K_M_* and *K_cat_* that are important in determining the rate-limiting step and inhibitor potency of enzymes.

Compared to the previously developed qFRET analyses that requires a standard curve, to relate detected fluorescent signal to concentrations of corresponding proteins, our new approach detects direct donor’s emission in real time instead of creating standard curves. According to the review of different qFRET analysis [26], more accurate and robust results can be obtained by observations using multi-channel instead of only one channel. In our approach, real-time fluorescent signal from both 475 nm and 530 nm were detected, while the previous approach only analyzed the fluorescent signal at 530 nm. Fluorescent signal can be readily related to protein concentration by a standard curve, but with some potential caveats. This includes that the fluorescent emissions may have variations for different batches of purified recombinant proteins, and pipetting errors and impure constitutes of protein can also affect the results. Moreover, the fitted standard curve cannot relate detected fluorescent signal exactly to the concentration of each sample.

Advantageously, the method we have developed in this study does not require expensive instrumentation, and instead the fluorescence intensity can be determined by general fluorescence spectroscopy or fluorescence plate readers that are widely available. Also, our approach only requires molecular cloning and protein expression steps, without radioactive labeling, and these fluorescently tagged proteins are in the aqueous phase, and this is closer to their normal environment in cells.

The potential application of FRET-based protease assay to characterize other SUMO-SENP pairs, as well as other proteases has been suggested previously [28]. In addition, our highly sensitive FRET-based assay can be potentially used in high-throughput biological assays, such as protease inhibitor screenings. The kinetic study can also be additionally used to characterize the properties of the inhibitors being screened (e.g., *Ki*, *IC50*). Thus, our highly sensitive qFRET-based protease assay method could provide a powerful new approach in developing genome-wide protease–substrate profiling and inhibitor screens.

## 5. Conclusions

The isopeptidase kinetics parameters of SENP1 were determined through a novel substrate preparation followed by a quantitative real time FRET assay and analysis. This represents the first case of isopeptidase kinetics determination and the methodology can be applied to determine other isopeptidase kinetics in general.

## Figures and Tables

**Figure 1 biomolecules-11-00673-f001:**
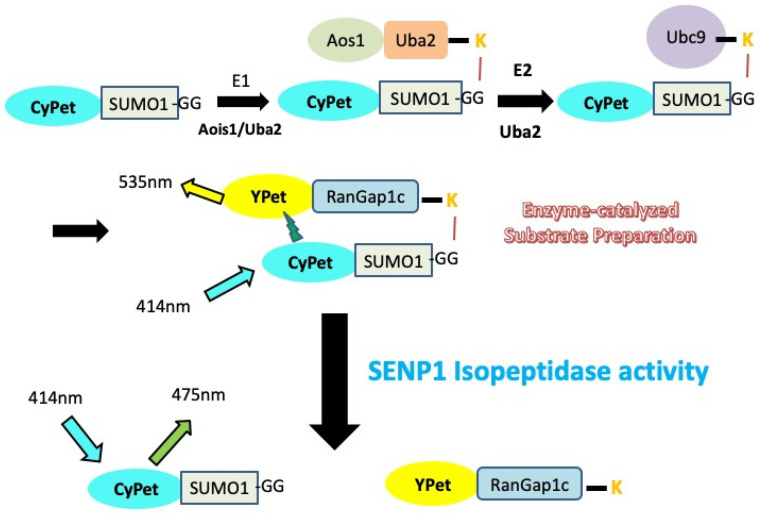
Principle of FRET-based protein assay to monitor the SUMO-RanGAP1C conjugation and deconjugation in vitro.

**Figure 2 biomolecules-11-00673-f002:**
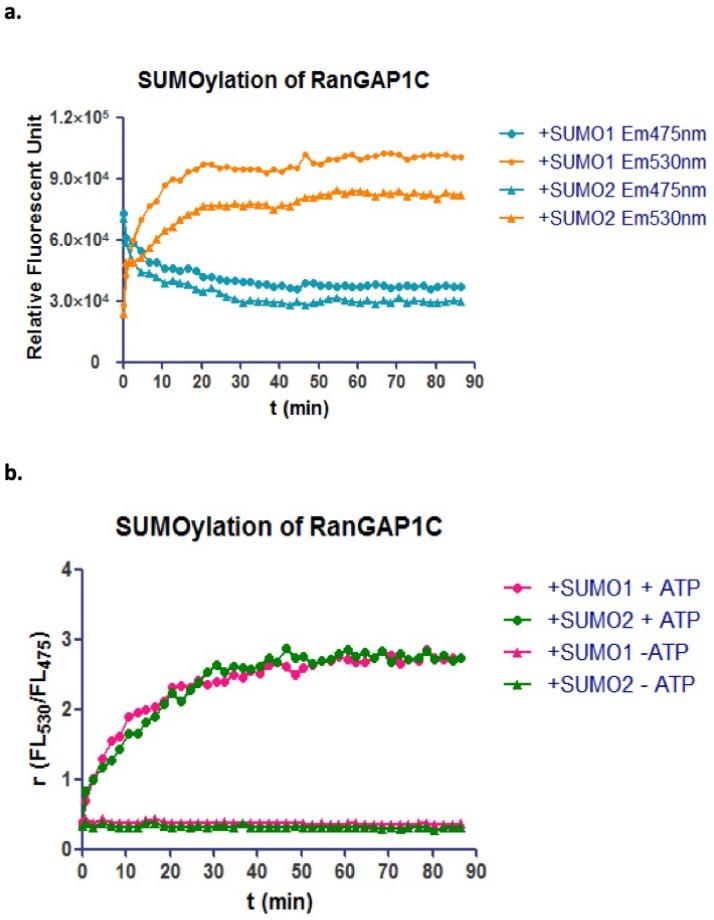
FRET-based protein assay to monitor the SUMO-RanGAP1C conjugation and deconjugation in vitro. (**a**) *Fluorescent* emission at 475 nm and 530 nm can be used to monitor SUMO1/2 conjugation under the excitation of 414 nm (with ATP). (**b**) Fluorescent emission ratio (*FL_530_/FL_475_*) of SUMO1/2 conjugation under the excitation of 414 nm (with/without ATP).

**Figure 3 biomolecules-11-00673-f003:**
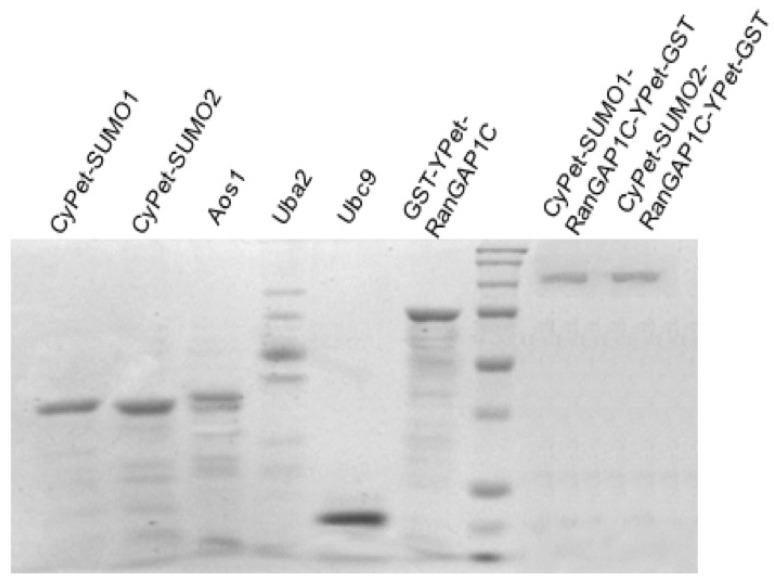
Coomassie staining of purified proteins in SUMO conjugation assay. From left to right: CyPet-SUMO1 (~40 kD) CyPet-SUMO2 (~40 kD), Aos1 (~38 kD), Uba2 (~65 kD), Ubc9 (~22 kD), GST-YPet-RanGAP1C (~75 kD), protein marker, CyPet-SUMO1-RanGAP1C-YPet-GST (~113 kD), and CyPet-SUMO2-RanGAP1C-YPet-GST (~113 kD).

**Figure 4 biomolecules-11-00673-f004:**
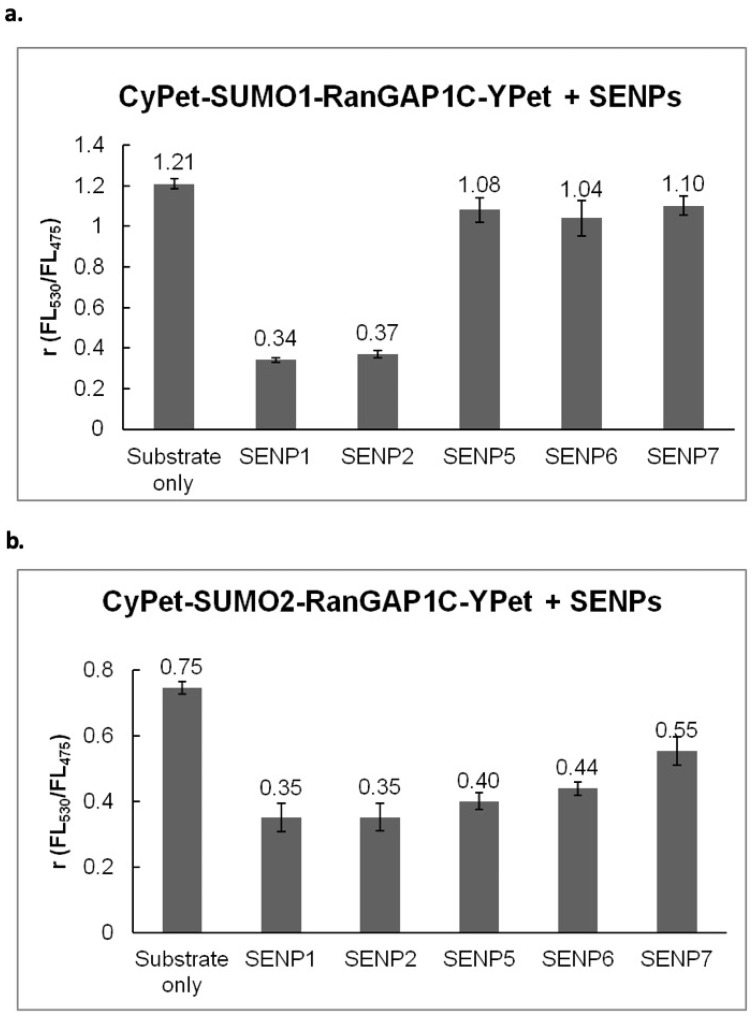
Characterization of SUMO1/2 deconjugation from RanGAP1C by SENP1/2/5/6/7C in developed FRET-based protease assay.

**Figure 5 biomolecules-11-00673-f005:**
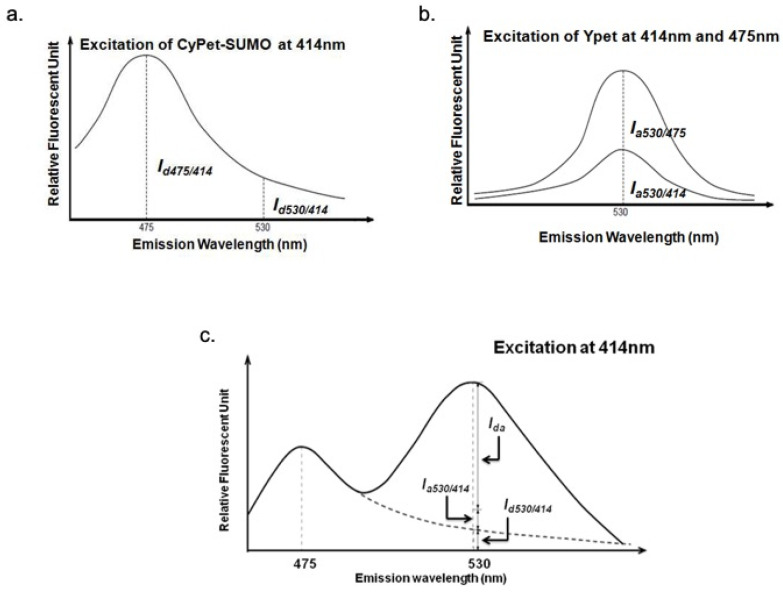
Quantitative analysis of fluorescent signals. (**a**) Determination of *α* factor using CyPet-SUMO1; (**b**) determination of *β* factor using YPet; (**c**) fluorescent emission at 530 nm at the excitation wavelength of 414 nm (*FL*_530/414_) can be divided into three components: FRET-induced YPet emission (*I_da_*), direct emission of unquenched CyPet (*I_d_*_530/414_), and direct emission of YPet (*I_a_*_530/414_).

**Figure 6 biomolecules-11-00673-f006:**
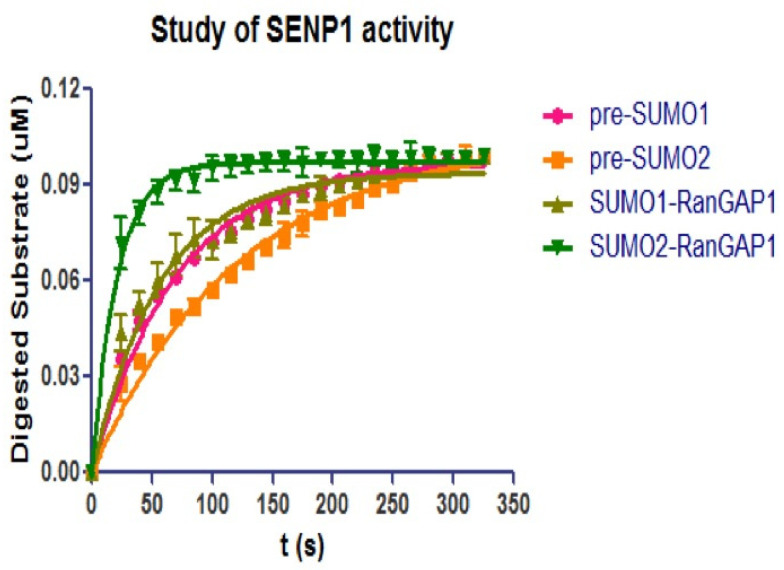
QFRET analysis to compare SENP1C’s endo- and iso-peptidase activities. 0.1 μM Substrate CyPet-(pre-SUMO1)-YPet, CyPet-(pre-SUMO2)-YPet, CyPet-SUMO1-RanGAP1C-YPet-GST, and CyPet-SUMO2-RanGAP1C-YPet-GST were incubated with 0.5 nM SENP1C in low salt Tris buffer at 37 °C. Reactions were monitored as the fluorescent emission at 475 nm and 530 nm (under the excitation of 414 nm) for every 10 s in the first 5 min. The digested substrate concentrations were calculated based on the developed qFRET analysis. Data were plotted in GraphPad Prism V and nonlinear regression.

**Figure 7 biomolecules-11-00673-f007:**
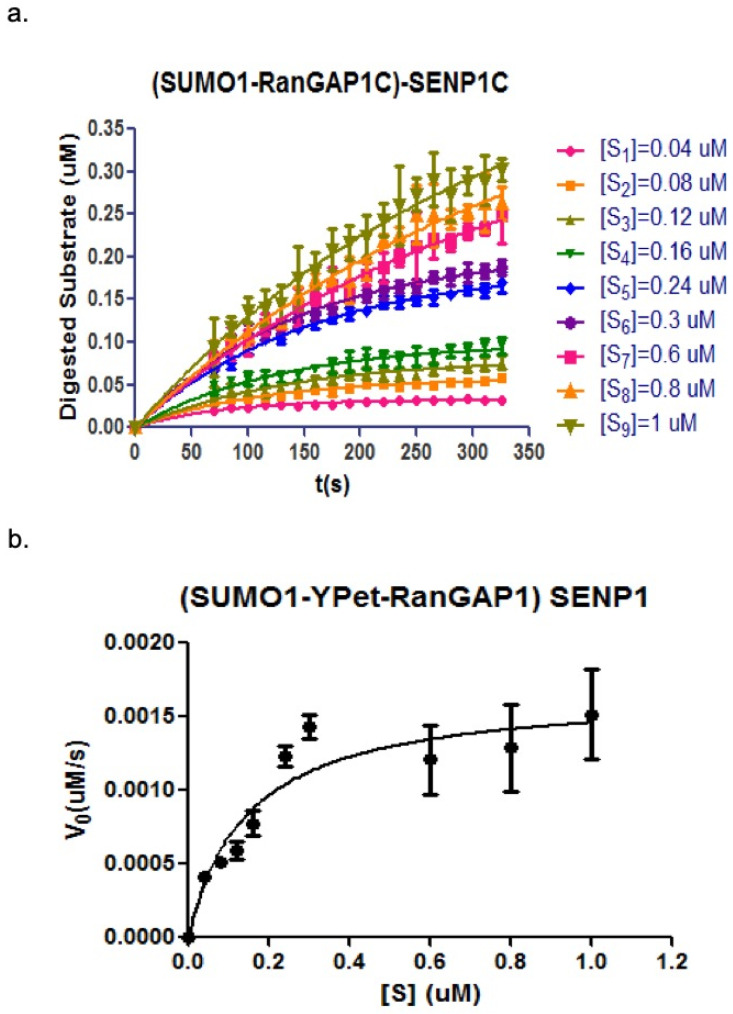
QFRET analysis in study the protease kinetics of SUMO1-RanGAP1C deconjugation by SENP1C. (**a**) Initial velocities determinations under different substrate concentrations. (**b**) Michaelis–Menten graphical analysis. Data were plotted and analyzed by GraphPad Prism 5 and nonlinear regression (one phase association model for (**a**) and Michaelis–Menten model for (**b**)).

**Table 1 biomolecules-11-00673-t001:** The initial velocity (*v*_0_) of pre-SUMO1/2 maturation and SUMO1/2- RanGAP1C deconjugation by SENP1C derived by qFRET analysis (substrate concentration was 0.1 μM).

SENP1 Substrate	Pre-SUMO1	Pre-SUMO2	SUMO1-RanGAP1	SUMO2-RanGAP1
*v*_0_ (×10^−3^ µM/s)	1.39 ± 0.03	0.89 ± 0.03	1.65 ± 0.07	4.60 ± 0.03

**Table 2 biomolecules-11-00673-t002:** The initial velocity (*v*_0_) of SUMO1-RanGAP1C deconjugation by SENP1C derived by qFRET analysis.

[CyPet-SUMO1-RanGAP1C-GST] (µM)	*v*_0_ (×10^−3^ µM/s)
0.04	0.41 ± 0.032
0.08	0.51 ± 0.038
0.12	0.59 ± 0.060
0.16	0.78 ± 0.081
0.24	1.23 ± 0.070
0.3	1.43 ± 0.081
0.6	1.21 ± 0.232
0.8	1.29 ± 0.295
1	1.52 ± 0.305

**Table 3 biomolecules-11-00673-t003:** Kinetic parameters of SUMO1-RanGAP1C deconjugation by SENP1C determined by qFRET analysis and compared to those derived from ratiometric FRET analysis [19].

Analysis Method	*K_M_* (µM)	*K_cat_* (s^−1^)	*K_cat_/K_M_* (×10^6^ M^−1^∙s^−1^)
QFRET analysis	0.14 ± 0.046	6.26 ± 0.63	43.47 ± 14.57
Ratiometric FRET analysis [19]	0.15 ± 0.015	8.27 ± 0.26	55.13 ± 5.78

## Data Availability

The data presented in this study are available on request from the corresponding author J.L.

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
