# Peer review of "Isopeptidase Kinetics Determination by a Real Time and Sensitive qFRET Approach"

_biomolecules, 2021, doi:10.3390/biom11050673_

Round 1

Reviewer 1 Report

The manuscript titled: Isopeptidase Kinetics Determination by A Real Time and Sensitive qFRET Approach by Liu et al, is an interesting work, addressing the specificity and kinetic of SENP activities in in vitro systems. The data are clearly described and the conclusions justified.   I suggest only a minor change in the introduction. The authors describe clearly the previous methodologies to measure deSUMOylation, and their limitations. However they only briefly mention their recently developed method in which they used an engineered FRET pair, CyPet and YPet that have a significantly improved FRET efficiency and fluorescent quantum yield,  the CyPet-(pre-SUMO)-YPet substrate. I would suggest to expand a bit on that to clarify how their method is superior to previously used FRET technology.   Another minor point. I noticed some comments, that look like note to self, in the text. I am assuming they were left from previous version of the manuscript. They should be removed.   In the results section, the authors claim that  CyPet-SUMO1-RanGAP1C-
YPet-GST and CyPet-SUMO2-RanGAPC-YPet-GST observed on the polyacrylamide gel show same molecular weight.
Based on this they conclude that there were no polySUMO2 chains conjugated to RanGAP1C (Fig. 3). I suggest the authors to run their sample on either a gradient gel or longer, in order to separate better the high molecular weight section of the gel. Also, maybe it is just the resolution of the image, but it appears that the background around those lanes is different, almost like they were from a different gel.  

Author Response

Dear editor and Reviewers,

Thank you very much for your invaluable and quickly comments and handling for our manuscript! I have carefully read the comments and revised the manuscript as following,

Reviewer 1

“The manuscript titled: Isopeptidase Kinetics Determination by A Real Time and Sensitive qFRET Approach by Liu et al, is an interesting work, addressing the specificity and kinetic of SENP activities in in vitro systems. The data are clearly described and the conclusions justified. I suggest only a minor change in the introduction. The authors describe clearly the previous methodologies to measure deSUMOylation, and their limitations. However they only briefly mention their recently developed method in which they used an engineered FRET pair, CyPet and YPet that have a significantly improved FRET efficiency and fluorescent quantum yield,  the CyPet-(pre-SUMO)-YPet substrate. I would suggest to expand a bit on that to clarify how their method is superior to previously used FRET technology.”

Thanks for the encouraging comments and invaluable suggestion to explain the advantages of our quantitative FRET method over other FRET methods! I have added several sentences as following to describe the advantages of our novel approach of quantitative FRET analysis method,

            “In these methods, we developed a novel quantitative FRET analysis method to extract the absolute FRET signal, which corresponds to un-digested substrate, CyPet-(pre-SUMO)-YPet, from the fluorescence signals directly from free donor, CyPet-(pre-SUMO), or acceptor, -YPet, from digested substrate. We used either a external standard curve of free donor CyPet, acceptor YPet, and FRET substrate, CyPet-(pre-SUMO)-YPet or an internal cross-correlation coefficiency method{Liu, 2012 #2248}{Liu, 2013 #895}{Jiang, 2013 #881}. Our approach can determine accurate digested or undigested substrates without interferences of free donor and acceptor in real time.”

“Another minor point. I noticed some comments, that look like note to self, in the text. I am assuming they were left from previous version of the manuscript. They should be removed.”  

Sorry for these! I have corrected these. I am not sure why this version was not the final one that I submitted.

“In the results section, the authors claim that CyPet-SUMO1-RanGAP1C-YPet-GST and CyPet-SUMO2-RanGAPC-YPet-GST observed on the polyacrylamide gel show same molecular weight. Based on this they conclude that there were no polySUMO2 chains conjugated to RanGAP1C (Fig. 3). I suggest the authors to run their sample on either a gradient gel or longer, in order to separate better the high molecular weight section of the gel. Also, maybe it is just the resolution of the image, but it appears that the background around those lanes is different, almost like they were from a different gel.”

            Thanks for the very careful comment! This gel picture was from one single picture. We did run low concentration gel before this one and did not see obvious poly-SUMOylated substrate. In addition, I think the mon-SUMOylated or poly-SUMOylated substrates does not affect the results. First, if there were some poly-SUMOylated substrate(s), it should be observed in this get as the two higher MW markers with >30kD could still be observed clearly above the mono-SUMOylated substrate. Second, we measured the major FRET signal which corresponds to the cleavage of the last CyPet-SUMO1 from YPet-RanGAP1c. The cleavage of the poly-SUMO substrate before the last one would be negligible from both concentration and distance to the acceptor. To include all the proteins in one gel, we had to run all the proteins in this concentration gel.

Sincerely,

Jiayu Liao

Reviewer 2 Report

In this manuscript, Liu et al describe an elegant qFRET-based  method for determining the enzymatic kinetics of the  isopeptidase SENP1, which is theoretically extensible also for analysis of other isopeptidases.  For this, they first  sumoylated  the  SENP1 substrate SUMO1/RanGAP1c generating an engineered FRET pair CyPet-SUMO1 and YPet-RanGAP1c. Then, by carrying out a qFRET analyses of real-time FRET signal  variation following SENP1-mediated  release of CyPet-SUMO1 and YPet-RanGAP1c from the conjugated substrate of CyPet-SUMO1/YPet-RanGAP1c they were able to obtain the kinetic parameters, Kcat, KM, and catalytic efficiency (Kcat/KM) of SENP1.

The manuscript by Liu et al. describes the development of  a novel protease assay  based on qFRET to determinate the isopeptidase kinetics of SENP1. There are several typos to be corrected, including the comment in lines 406-408.

Author Response

Dear editor and Reviewers,

Thank you very much for your invaluable and quickly comments and handling for our manuscript! I have carefully read the comments and revised the manuscript as following,

Reviewer 2

“In this manuscript, Liu et al describe an elegant qFRET-based  method for determining the enzymatic kinetics of the  isopeptidase SENP1, which is theoretically extensible also for analysis of other isopeptidases.  For this, they first sumoylated  the  SENP1 substrate SUMO1/RanGAP1c generating an engineered FRET pair CyPet-SUMO1 and YPet-RanGAP1c. Then, by carrying out a qFRET analyses of real-time FRET signal  variation following SENP1-mediated  release of CyPet-SUMO1 and YPet-RanGAP1c from the conjugated substrate of CyPet-SUMO1/YPet-RanGAP1c they were able to obtain the kinetic parameters, Kcat, KM, and catalytic efficiency (Kcat/KM) of SENP1.

The manuscript by Liu et al. describes the development of  a novel protease assay  based on qFRET to determinate the isopeptidase kinetics of SENP1. There are several typos to be corrected, including the comment in lines 406-408.”

            Thanks! I am sorry for this! These have been corrected. I am not sure why this version was not the final version that was submitted.

Sincerely,

Jiayu Liao

This manuscript is a resubmission of an earlier submission. The following is a list of the peer review reports and author responses from that submission.